# Patch-Wise Point Cloud Generation: A Divide-and-Conquer Approach

## Abstract

A generative model for high-fidelity point cloud is of great importance in synthesizing 3d environments for applications such as autonomous driving and robotics. Despite recent success of deep generative models for 2d images, it is non-trivial to generate point cloud without a comprehensive understanding on both local and global geometric structures. In this paper, we devise a new 3d point cloud generation framework using a divide-and-conquer approach, where the whole generation process can be divided into a set of patch-wise generation tasks. Specifically, all patch generators are based on learnable priors which aims to capture the information of geometry primitives. We introduce point- and patch-wise transformers to enable the interactions between points and patches. Therefore, the proposed divide-and-conquer approach contributes to a new understanding on point cloud generation from the geometry constitution of 3d shapes. Experimental results on a variety of object categories from the most popular point cloud dataset, ShapeNet, show the effectiveness of the proposed patch-wise point cloud generation, where it clearly outperforms recent state-of-the-art methods for high-fidelity point cloud generation.

## 1 Introduction

With the rapid development of depth sensing and laser scanning technologies (Zhang, 2012; Keselman et al., 2017), 3d data has become more and more popular for modeling scene and objects in real-world environments, especially in the applications such as autonomous driving (Nagy & Benedek, 2018) and virtual reality (Wirth et al., 2019). As a standard 3d acquisition format, point cloud is a compact geometric representation, which is simple and useful for understanding geometric shape structures of complex objects and large-scale scenes in real-world applications. However, it is non-trivial to collect and annotate large-scale point clouds in many real-world applications. Therefore, with recent development of deep generative models (Goodfellow et al., 2014), point cloud generation has received increasing attention from the community, especially in the related point cloud tasks such as shape completion (Huang et al., 2020; Cai et al., 2022) and synthesis (Achlioptas et al., 2018; Tang et al., 2022; Hoogeboom et al., 2022).

Recently, deep generative models have shown great success in generating realistic 2d images from complex distributions by using generative adversarial networks (GANs) (Goodfellow et al., 2014) and/or variational autoencoders (VAEs) (Kingma & Welling, 2014). Though remarkable progress has been made in generating 2d images, point cloud generation still remains very challenging due to the irregular sampling patterns in 3d space, especially for complex object shapes/structures (Achlioptas et al., 2018; Valsesia et al., 2018; Shu et al., 2019; Yang et al., 2019; Hui et al., 2020b; Kim et al., 2020; Klokov et al., 2020; Wen et al., 2021; Li et al., 2021; Luo & Hu, 2021; Zhang et al., 2021; Tang et al., 2022). Motivated by that complex shapes can be also composed from a set of geometry primitives (or meta shapes), we thus propose to explore the great potential of patch-wise point cloud generation using a divide-and-conquer approach. We show an intuitive example of patch-wise point cloud generation in Fig. 1. Specifically, the overall point cloud generation process is first divided into a set of patch generation tasks, where the learnable 2d patch priors are used to generate point cloud patches. After that, those generated patches are combined into a single point cloud as the final point cloud. The intuition behind such a patch-wise generation is that we consider the 3d shape to be constructed by a set of geometry primitives learned by deep neural networks.

To jointly learn patch priors and generate point clouds, the overall patch-wise point cloud generation framework follows the widely used VAE-GAN (Larsen et al., 2016). To learn patch priors, each patch prior is a set of 2d points which are randomly initialized and then learned by deep neural networks during training. With a random latent code, the patch generator transforms each patch prior into a set of 3d points (or a point cloud patch), and the union of these 3d point sets will form the target point cloud. Specifically, during training, the input of each patch generator is a concatenation of the latent representation of training sample and the 2d patch prior. During testing, we use the concatenation of a randomly sampled latent code and the learned 2d patch prior as the input of each patch generator. Furthermore, to develop effective patch generators by enabling the information flow in different

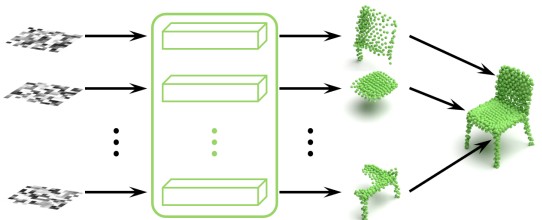

Figure 1: An intuitive example of patch-wise point cloud generation using a divide-and-conquer approach. Specifically, a set of patch generators learn to transform 2d patch priors into a set of point cloud patches, which are part of the target 3d shape.

patches, we introduce transformers to patch-wise point cloud generation to explore local and global relationships, i.e., different points in each patch and different patches. Specifically, we explore three different types of patch generators: 1) the mlp generator (or MLP-G) as the baseline; 2) the transformer generator (or PointTrans-G) which uses transformer to explore point-wise relationship in each patch; and 3) the dual-transformer generator (or DualTrans-G), where transformers are used to explore both point-wise and patch-wise relationships.

Previous works (Groueix et al., 2018; Deprelle et al., 2019; Williams et al., 2019) also adopt the multi-patch strategy, but all of them focus on shape analysis tasks (i.e., reconstruction and matching) rather than generation. For example, AtlasNet (Groueix et al., 2018) is always compared with other works (Luo & Hu, 2021) to evaluate the reconstruction quality of auto-encoders. By contrast, our patch-wise method is actually based on point clouds, and we use a VAE-GAN in this paper as our baseline. In this paper, our main contributions can be summarized as follows:

1) We introduce a divide-and-conquer approach for point cloud generation, where the point cloud is constructed from a set of patches learned in a patch-wise manner.

2) We devise three different types of generator modules using transformers to explore both point- and patch-wise relationships for point cloud generation.

3) We conduct extensive qualitative and quantitative experiments on the most popular point cloud generation dataset, ShapeNet (Chang et al., 2015), to show the effectiveness of the proposed method as well as its key components.

## 2 Related Work

**Deep Learning for 3D Point Cloud** Learning a robust deep representation from point cloud is of great importance for point cloud understanding (Qi et al., 2017a; Ren et al., 2022; Chu et al., 2022). Among recent methods, PointNet is a pioneering pointwise CNN (Qi et al., 2017a), while several following hierarchical architectures have been further developed to better capture local structures, such as PointNet++ (Qi et al., 2017b), PointCNN (Li et al., 2018b), PointConv (Wu et al., 2019), and KPConv (Thomas et al., 2019). Specifically, PointCNN (Li et al., 2018b) also introduce a new $\chi$-conv transformation which transforms points into a latent canonical order and then applies convolutional operator. Given the irregular and unordered nature of point clouds, recent success of transformer architectures provides a promising mechanism to encode rich relationships between points (Vaswani et al., 2017). Inspired by this, various transformer architectures have been recently proposed for point cloud analysis (Zhao et al., 2021; Guo et al., 2021; Mazur & Lempitsky, 2021; Pan et al., 2021; Yang et al., 2022). Specifically, Zhao et al. (2021) propose to apply self-attention in the local neighborhood of each point, where the proposed transformer layer is invariant to the permutation of the point set, making it suitable for point set processing tasks. Guo et al. (2021) propose a novel point

cloud transformer framework or PCT to replace the original self-attention module with a more suitable offset-attention module, which includes an implicit Laplace operator and a normalization refinement. Mazur & Lempitsky (2021) introduce a new building block, which combines the ideas of spatial transformers and multi-view convolutional networks with the efficiency of standard convolutional layers in two dense grids. The new block operates via multiple parallel heads, whereas each head rasterizes feature representations of individual points into a low-dimensional space in a differentiable way, and then uses dense convolution to propagate information across points.

**Point Cloud Generation** In the past few years, deep generative models have been explored for point cloud analysis. Specifically, Achlioptas et al. (2018) first explore different GAN architectures in both raw data space and latent space with a pretrained autoencoder, which has become a popular baseline for point cloud generation. Then, Shu et al. (2019) adopt GAN with a tree structure, namely TreeGAN, to preserve ancestor information instead of neighbor information to generate new points. To enhance the generation quality, Hui et al. (2020a) and Wen et al. (2021) further divide a difficult task of generating high-fidelity sample into multiple steps, which significantly improves point cloud generation performance. In addition, VAE also attracts increasing attention in point cloud generation due to its elegant formulation and promising performance. For example, Yang et al. (2019) propose a probabilistic VAE to generate point cloud by modeling it as a two-level hierarchical distribution, but it converges slowly and usually fails in cases with thin structures. Kim et al. (2021) adopt attention-based set transformers (Lee et al., 2019) into the VAE framework, and extend it to a hierarchy of latent variables to account for flexible subset structures. Probabilistic model is also a popular method for point cloud generation. Specifically, Luo & Hu (2021) learn the reverse diffusion process that transforms the noise distribution to the desired shape distribution. Klokov et al. (2020) introduce a latent variable model using normalizing flows with affine coupling layers to generate 3d point clouds of an arbitrary size given a latent shape representation. Zhang et al. (2021) propose a Markov chain based 3D generative model that iteratively mends the shape to a learned distribution. Recently, Tang et al. (2022) propose the WarpingGAN to formulate the generation process as the learning of a function that warps multiple 3D priors into different local regions.

## 3 Preliminaries

In this section, we provide background knowledge for deep generative models and transformer architectures.

**VAE-GAN** Given a training set $X = \{x_i\}_{i=1}^n$, we assume $x_i$ is sampled from a generative process $p(x|z)$, where $z$ refers to the latent variable. The objective of VAE (Kingma & Welling, 2014) is to simultaneously train an inference network (or encoder) $q_\phi(z|x)$ and a generator network (or decoder) $p_\psi(x|z)$. Therefore, the VAE model can be trained by jointly minimizing the negative of evidence lower bound (ELBO),

$$\mathcal{L}_{ELBO}(\phi, \psi; x) = -\mathbb{E}_{z \sim q_\phi(z|x)}[log(p_\psi(x|z))] + KL[q_\phi(z|x)||p(z)], \tag{1}$$

where $p(z)$ is the prior distribution, i.e., $\mathcal{N}(0, I)$, and $KL$ indicates the Kullback–Leibler divergence. The first term $-\mathbb{E}_{z \sim q_\phi(z|x)}[log(p_\psi(x|z))]$ can be reduced to a standard point-wise reconstruction loss and the second term is the regularization term to prevent the conditional $q_\phi(z|x)$ from deviating from the Gaussian prior $\mathcal{N}(0, I)$. Lastly, the whole network are jointly optimized by

$$\min_{\phi, \psi} \mathbb{E}_{x \sim p_{data}(x)} \mathcal{L}_{ELBO}(\phi, \psi; x), \tag{2}$$

where $p_{data}$ is the distribution induced by the training set $X$. The objective of VAE-GAN (Larsen et al., 2016) augments Eq. (2) with the GAN objective. Specifically, the modified ELBO computes the reconstruction loss in the feature space of the discriminator instead of the sample space:

$$\mathcal{L}_{ELBO}(\phi, \psi, D; x) = \mathbb{E}_{z \sim q_\phi(z|x)}[||F_D(x) - F_D(G(z))||_2^2] + KL[q_\phi(z|x)||p(z)], \tag{3}$$

where $D$ is the discriminator, $G$ is the generator and $F_D(\cdot)$ denotes the feature embedding from the discriminator. The modified GAN objective considers both reconstructed point clouds (latent code from $q_\phi(z|x)$)

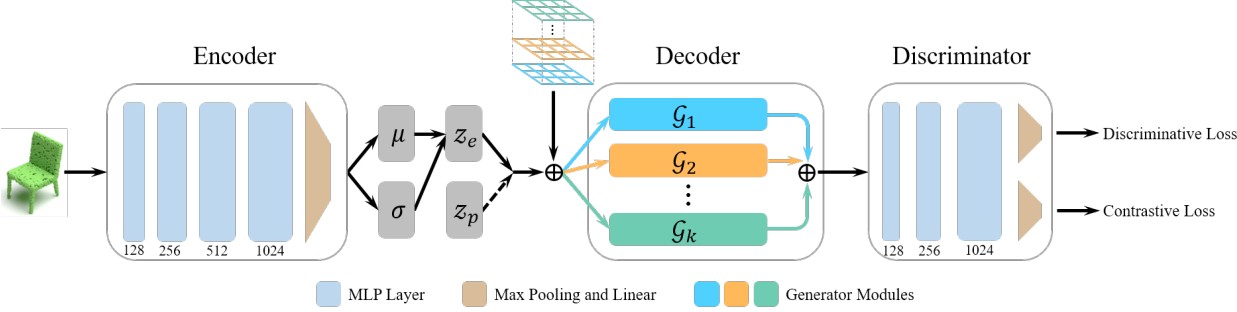

Figure 2: Overview of the main framework. For we adopt the VAE-GAN framework (Larsen et al., 2016), thus there are three components in our network: encoder, decoder and discriminator. The decoder consists of at least one generator module $\mathcal{G}_i$ ($1 \leq i \leq k$). $z_e$ is computed by the encoder and $z_p$ is sampled from prior distribution. The key innovation of our method is that we deform multiple patches to produce point clouds, which is referred to as the divide-and-conquer generation.

and sampled point clouds (latent code from the prior $p(z)$) as its fake samples:

$$\mathcal{L}_{GAN}(\phi, \psi, D; x) = \mathbb{E}_{z \sim q_\phi(z|x)} log(1 - D(G(z))) + \mathbb{E}_{z \sim p(z)} log(1 - D(G(z))) + log D(x). \tag{4}$$

Overall, the final objective becomes:

$$\min_{\phi, \psi} \max_{D} \mathcal{L}_{ELBO}(\phi, \psi, D; x) + \mathcal{L}_{GAN}. \tag{5}$$

**Transformer** A vanilla transformer (Vaswani et al., 2017; Dosovitskiy et al., 2020) consists of an encoder module and a decoder module. Each encoder/decoder layer is composed of a multi-head self-attention layer and a position-wise feed-forward network. Specifically, each self-attention layer in transformer adopts a "query-key-value" mechanism as follows. The input feature sequence is first transformed into three different vectors, queries $Q \in R^{N \times d_q}$, keys $K \in R^{N \times d_k}$ and values $V \in R^{N \times d_v}$, where $N$ is the length of feature sequence and $d_q, d_k, d_v$ denote the dimensions of keys, queries and values, respectively. Then, the scaled dot-product attention used in transformer can be formulated as follows:

$$Attention(Q, K, V) = softmax(\frac{Q \cdot K^T}{\sqrt{d_k}}) \cdot V. \tag{6}$$

## 4 Method

In this section, we introduce the proposed patch-wise point cloud generation. Specifically, we first present an overview of our network, and then describe each component in the proposed method in detail.

### 4.1 Overview

As mentioned in the previous section, we adopt the VAE-GAN (Larsen et al., 2016) framework, and our model consists of three components, one encoder, one decoder and one discriminator. The main VAE-GAN framework for patch-wise point cloud generation is shown in Fig. 2. It also can be considered as a VAE network equipped with a discriminative network, in which the VAE component trains the decoder to reconstruct real samples with plausible variation, while the discriminative module enforces the decoder to produce realistic point clouds from prior distribution. Different from typical VAE-GAN models, we have a set of different patch generators, which is based on the learnable patch priors. The output of all patch generators are combined to generate the final point cloud. We introduce the patch-wise generation in a divide-and-conquer manner in next section.

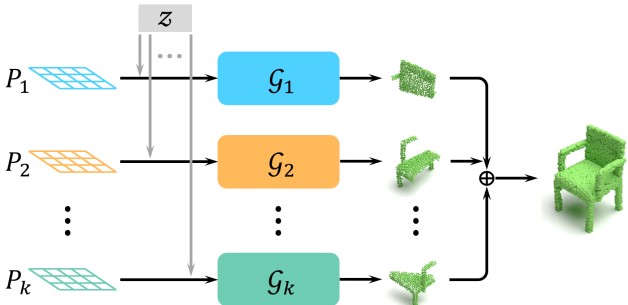

Figure 3: An overview of the proposed patch-wise generation. The input to each generator module $\{\mathcal{G}_i\}, 1 \leq i \leq k$ is the concatenation of the latent representation $z$ and the patches $\{\mathcal{P}_i\}, 1 \leq i \leq k$. The $k$ deformed patches make up the final 3d shape.

## 4.2 Patch-wise Generation

**Divide**  Generative models usually take a latent code $z$ as the input to produce realistic samples. However, this generation process fails to explore the geometry constitution of point cloud. Therefore, we investigate point cloud generation using a divide-and-conquer strategy. Specifically, for the task of point cloud generation, our goal is to generate the target shapes using a set of $k$ learned patch priors $\mathcal{P}_1, \mathcal{P}_2, ..., \mathcal{P}_k$, which are then transformed via generator modules $\mathcal{G}_1, \mathcal{G}_2, ..., \mathcal{G}_k$. Given the training point cloud, we represent each input point cloud by a feature vector $z$ (i.e., either $z_e$ computed by the encoder or $z_p$ sampled from prior distribution). Each patch generator $\mathcal{G}_i$ ($1 \leq i \leq k$) takes the feature vector $z$ and the patch priors $\mathcal{P}_i$ ($1 \leq i \leq k$) as inputs, and generate the corresponding 3d points for the target shape.

**Combine**  The output point cloud $O$ can thus be formed as the union of the generated patches of point cloud, i.e.,

$$O = \cup_{i=1}^{k} \mathcal{G}_i(z, \mathcal{P}_i). \tag{7}$$

However, simply assembling the generated patches may not always lead to satisfying point clouds. Therefore, the point-wise reconstruction term in Eq. (1) imposes a constraint on the final generation and contribute to producing desirable results. Specifically, we train the network by minimizing the Chamfer distance (Fan et al., 2017) between the $O$ and training data $x_i \in X$. The patch-wise generators are shown in Fig. 3. We aim to automatically learn the patch priors $\{\mathcal{P}_i, 1 \leq i \leq k\}$ over a training collection. The intuition behind the approach is that if the patch priors $\{\mathcal{P}_i, 1 \leq i \leq k\}$ have useful geometry information to reconstruct the target, the patch generators $\{\mathcal{G}_i, 1 \leq i \leq k\}$ should be easier to learn and more interpretable. The patch priors $\{\mathcal{P}_i, 1 \leq i \leq k\}$ are learned over the entire training set and do not depend on the input during testing. That is, at the testing time, we take the patch priors $\{\mathcal{P}_i, 1 \leq i \leq k\}$ and a randomly sampled latent vector as the input of the patch generators $\{\mathcal{G}_i, 1 \leq i \leq k\}$ to generate the output 3d point cloud.

## 4.3 Network Structures

In this subsection, we introduce the detailed structures of the encoder, decoder, and discriminator.

**Encoder**  We use a simplified version of the PointNet (Qi et al., 2017a) as the encoder. Specifically, we utilize multi-layer perceptron (MLP) layers to learn both low- and high-level representations in a pointwise manner. Given the training data with $N$ points, the input and output of each MLP layer are point features with size $N \times C_{in}$ and $N \times C_{out}$, where $C_{in}$ and $C_{out}$ denote the numbers of channels in the input and output feature attributes, respectively. In our setting, we represent each 3d point of the training data as a 1024 dimensional vector using three hidden MLP layers of 128, 256 and 512 neurons with the ReLU activation function. We then apply a max-pooling layer over all point features followed by a fully connected layer, producing a global shape feature used as input to the patch generators.

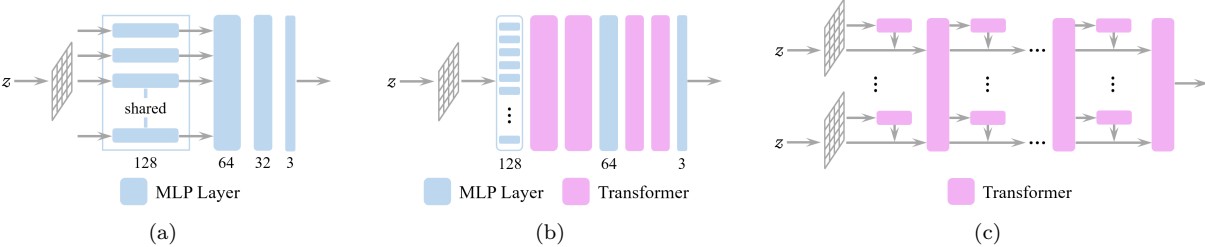

Figure 4: Three different types of generator modules: (a) MLP-G, (b) PointTrans-G and (c) DualTrans-G.

**Decoder** Patch generators $\{\mathcal{G}_i, 1 \leq i \leq k\}$ aim to synthesize point clouds by using patch priors $\mathcal{P}_i$ ($1 \leq i \leq k$). The deformation from patch priors to point cloud is not difficult, while we can expect the quality of generation to increase using more complex generators. As mentioned before, each patch is a set of 2d points with size $N_k$, where $\sum N_k = N$. The input to each generator module $\mathcal{G}_i$ ($1 \leq i \leq k$) is a feature vector with size $N_k \times (d_z + d_{patch})$, where $d_z$ and $d_{patch}$ are the dimensions of latent code and the patch priors, respectively. We always use two dimensional patch priors, i.e., $d_{patch} = 2$. The output of the patch generator is with the size $N_k \times 3$. The final output point cloud is the union of the $k$ outputs of the generator modules. In this paper, we explore three types of generator modules: MLP generator (MLP-G), point transformer generator (PointTrans-G) and point-patch transformer generator (DualTrans-G), as shown in Fig. 4. The last two generator modules incorporate the popular transformer structure. Transformer in PointTrans-G is called pointwise transformer and in DualTrans-G called patchwise transformer. The two types of transformer have the same architecture but used to extract point-point and patch-patch relationships, respectively. We introduce the details of different generators as follows:

1) MLP-G: this generator is built on MLP layers, which takes as input the concatenation of the latent code and the coordinates of 2d points from the associated patch. The output is a collection of learned 3d points which will be used to construct the 3d shape. The MLP layer is the same architecture as the one used in the encoder but with different dimensions.

2) PointTrans-G: in the MLP-G, each point in the patch is deformed in a pointwise manner. There is no information exchange between different points. Inspired by the recent transformer models on vision tasks, here we insert pointwise transformer layers between MLP layers to extract local features by aggregating the information of neighborhood points. Different from the vanilla transformer (Vaswani et al., 2017), the transformer used in our work only contains the encoder module.

3) DualTrans-G: based on the PointTrans-G, we extend the usage of transformer between different patches. Specifically, patches are first fed into the pointwise transformer layer and then the patchwise transformer layer. Thus there is information flow between different patches, as well as different points on local patch. We call this structure DualTrans because it allows to learn both local and global information.

Note that, in MLP-G and PointTrans-G, each patch associates with a generator module and patches are transformed one by one. While in DualTrans-G, all patches are feed into the generator module simultaneously. These three types of generator modules are shown in Fig. 4.

**Discriminator** In adversarial learning, a discriminator distinguishes whether the input is produced by the generator or sampled from the ground truth distribution, while the generator aims to fool the discriminator by generating realistic samples. In this paper, we use both discriminative and contrastive features to boost the generator for high-fidelity point cloud generation. Specifically, we first employ a feature extraction backbone with the MLP layers. We then apply max-pooling over point features followed by two different branches, a discriminative one and a contrastive one, aiming to generate discriminative and contrastive shape features.

### 4.4 Optimization

Though the objective in Eq. (5) improves the overall quality, it can not accurately explore instance-level fidelity. Inspired by (Oord et al., 2018), we adopt a similar contrastive learning loss to further enhance the patch-wise generation process as follows:

$$\mathcal{L}_{con}(\phi, \psi, D; \hat{X}) = -\mathbb{E}_{\hat{X}}[log \frac{f(\hat{x}_i, aug(\hat{x}_i))}{f(\hat{x}_i, aug(\hat{x}_i)) + \sum\limits_{1 \leq i,j \leq n, i \neq j} f(\hat{x}_i, \hat{x}_j)}], \tag{8}$$

where $\hat{X}$ is the output of the decoder. The latent code fed into the decoder can be either $z_e$ computed by the encoder or the $z_p$ sampled from prior distribution, and $\hat{x}_i, \hat{x}_j \in \hat{X}$. In Eq. (8), $f(\cdot, \cdot) = e^{h(F_D(\cdot), F_D(\cdot))}$, where $h(\cdot, \cdot)$ is the cosine similarity function that measures compatibility between feature embeddings, and $F_D(\cdot)$ denotes the feature embeddings from the discriminator $D$. The positive pair comprises instance $\hat{x}_i$ and its augmentation $aug(\hat{x}_i)$, such as rotating and jittering, and the negative pair consists of different instance tuples $(\hat{x}_i, \hat{x}_j)_{i \neq j}$ in the same batch. Therefore, the overall learning objective is,

$$\min\limits_{\phi, \ \psi} \max\limits_{D} \ \mathcal{L}_{ELBO}(\psi, \phi, D; x) + \mathcal{L}_{GAN} + \mathcal{L}_{con}. \tag{9}$$

To better understand the role of contrastive loss, we first explain the discriminative loss, which encourages generated samples to be close to real samples and far away from fake samples. From this aspect, it can be regarded as the "class-level contrastive loss" (real samples for positive pairs and fake samples for negative pairs). However, it does not measure the instance-level fidelity among results in the same batch. Therefore, the motivation behind the contrastive loss in our paper is to promote each generated sample to be as close as possible to its augmented one while being different from all other samples in current batch. In this way, the contrastive loss brings a significant improvement, which is one of the major differences comparing with Atlasnet (Groueix et al., 2018; Deprelle et al., 2019). The contrastive loss encourages each instance to be different with the other samples in the sample batch, and thus leading to high-fidelity generation.

## 5 Experiments

In this section, we first introduce implementation details, including the dataset and hyper-parameters, and then present experimental results. We compare our method with recent state-of-the-art point cloud generation methods and perform ablation studies on its important components.

### 5.1 Implementation Details

We sample points uniformly on mesh objects from the ShapeNet (Chang et al., 2015) dataset. In our experiments, we choose three widely-used object categories, chair, car and airplane, and each category with 2048 points. We use 8 patches for point cloud generation, and the dimension of latent code $z_e$ and $z_p$ is 128. We implement the method using PyTorch. All our models are trained on NVIDIA GeForce RTX 2080Ti GPU. As we introduce three different types of generator modules, each experiment is conducted on all of them. The sizes of MLP layers used in MLP-G and PointTrans-G are listed in Fig. 4. Here, we give the parameters of the transformer layers in PointTrans-G and DualTrans-G. As mentioned before, there are two types of transformer layers, pointwise transformer and patchwise transformer, in our architecture. Although they are used to extract different features, they have the same structure just with different parameters. Both types of transformers only contain one encoder module. We use 4-head self-attention pointwise transformer and 8-head self-attention patchwise transformer. The output channels of pointwise transformer layer in PointTrans-G keep the same as the MLP layer before it. In DualTrans-G, the output channels of pointwise transformer layer and patchwise transformer layer are 512 and 1024, respectively. We stack four interleaving structures of pointwise transformer and patchwise transformer as shown in Fig. 4.

| Class | Model | JSD(↓) | MMD(↓) | | COV(↑, %) | | 1-NNA(↓, %) | |
|---|---|---|---|---|---|---|---|---|
| | | | CD | EMD | CD | EMD | CD | EMD |
| Chair | raw-GAN | 11.5 | 2.57 | 12.8 | 33.99 | 9.97 | 71.75 | 99.47 |
| | latent-GAN | 4.59 | 2.46 | 8.91 | 41.39 | 25.68 | 64.43 | 85.27 |
| | PC-GAN | 3.90 | 2.75 | 8.20 | 36.50 | 38.98 | 76.03 | 78.37 |
| | WarpingGAN | - | - | 8.7 | - | **53.75** | - | - |
| | PointFlow | **1.74** | 2.42 | 7.87 | 46.83 | 46.98 | 60.88 | 59.89 |
| | PVD | 1.78 | 2.45 | 7.73 | 47.19 | 47.13 | 58.44 | **55.76** |
| | DPC | 1.80 | 2.58 | 7.78 | 48.94 | 47.52 | 60.11 | 69.06 |
| | SetVAE | - | 2.55 | 7.82 | 46.98 | 45.01 | 58.76 | 61.48 |
| | Ours (MLP-G) | 2.14 | 2.45 | 7.83 | 47.21 | 46.52 | 59.90 | 60.31 |
| | Ours (PointTrans-G) | 1.92 | 2.47 | 7.75 | 47.17 | 46.37 | 58.55 | 60.74 |
| | Ours (DualTrans-G) | 1.87 | **2.37** | **7.69** | **47.30** | 46.63 | **57.82** | 60.06 |
| Car | raw-GAN | 12.8 | 1.27 | 8.74 | 15.06 | 9.38 | 97.87 | 99.86 |
| | latent-GAN | 4.43 | 1.55 | 6.25 | 38.64 | 18.47 | 63.07 | 88.07 |
| | PC-GAN | 5.85 | 1.12 | 5.83 | 23.56 | 30.29 | 92.19 | 90.87 |
| | WarpingGAN | - | - | - | - | - | - | - |
| | PointFlow | 0.87 | 0.91 | 5.22 | 44.03 | 46.59 | 60.65 | 62.36 |
| | PVD | 0.84 | 0.87 | 5.10 | 47.94 | 46.70 | **57.15** | **56.44** |
| | DPC | - | - | - | - | - | - | - |
| | SetVAE | - | 0.88 | **5.05** | 48.58 | 44.60 | 59.66 | 63.35 |
| | Ours (MLP-G) | 0.83 | 0.87 | 5.17 | 48.75 | **46.81** | 60.34 | 61.95 |
| | Ours (PointTrans-G) | 0.80 | 0.87 | 5.24 | 48.30 | 46.22 | 59.81 | 62.26 |
| | Ours (DualTrans-G) | **0.78** | **0.85** | 5.14 | **49.12** | 46.54 | 59.60 | 61.81 |
| Airplane | raw-GAN | 7.44 | 0.261 | 5.47 | 42.72 | 18.02 | 93.58 | 99.51 |
| | latent-GAN | 4.62 | 0.239 | 4.27 | 43.21 | 21.23 | 86.30 | 97.28 |
| | PC-GAN | 4.63 | 0.287 | 3.57 | 36.46 | 40.94 | 94.35 | 92.32 |
| | WarpingGAN | - | - | 3.3 | - | 48.75 | - | - |
| | PointFlow | 4.92 | 0.217 | 3.24 | **46.91** | 48.40 | 75.68 | 75.06 |
| | PVD | 4.77 | 0.225 | 3.18 | 46.77 | 48.42 | 75.32 | **70.57** |
| | DPC | 4.67 | 0.218 | 3.06 | 48.71 | 45.47 | 64.83 | 75.12 |
| | SetVAE | - | 0.199 | 3.07 | 43.45 | 44.93 | 75.31 | 77.65 |
| | Ours (MLP-G) | 4.58 | 0.236 | 3.04 | 44.30 | 49.10 | 76.90 | 76.11 |
| | Ours (PointTrans-G) | **4.57** | 0.186 | 3.12 | 45.87 | 48.58 | 75.21 | 75.40 |
| | Ours (DualTrans-G) | 4.61 | **0.180** | **3.04** | 45.55 | **49.22** | **74.97** | 75.31 |

Table 1: Quantitative comparisons with recent state-of-the-art methods. ↑: the higher the better, ↓: the lower the better. The best scores are highlighted in bold. JSD scores and MMD-EMD scores are multiplied by $10^2$. MMD-CD scores are multiplied by $10^3$.

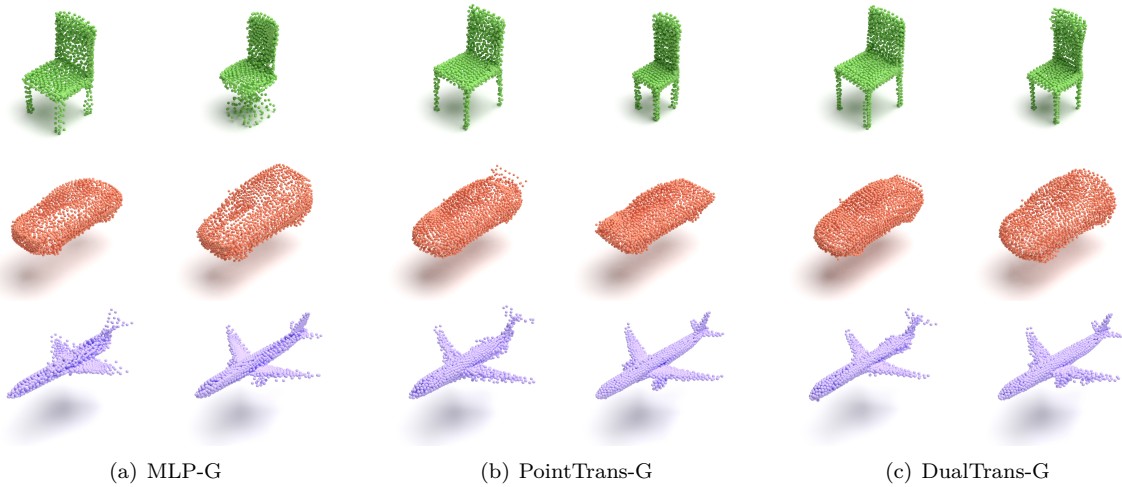

(a) MLP-G        (b) PointTrans-G        (c) DualTrans-G

Figure 5: Examples of point cloud generated by our model. From top to bottom: chair, car and airplane.

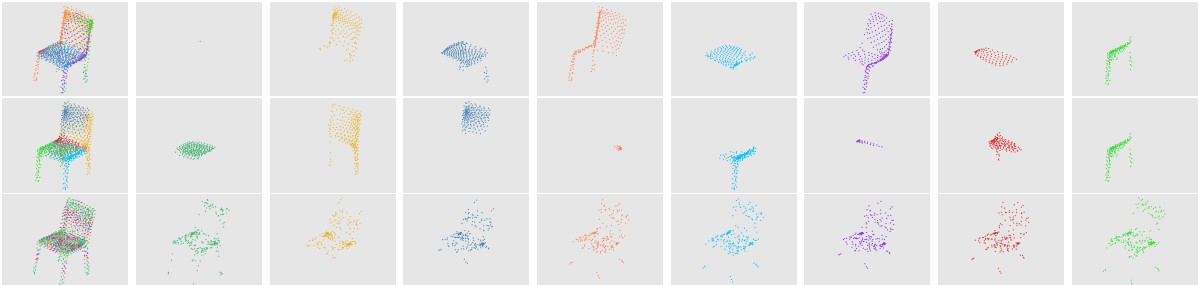

Figure 6: Generated point cloud and the learned patches. From top to bottom: MLP-G, PointTrans-G and DualTrans-G. The first column is the 3d shape and the rest columns are different patches.

## 5.2 Comparison with Recent State-of-the-Arts

We compare the proposed method with recent state-of-the-art methods, including raw-GAN (Achlioptas et al., 2018), latent-GAN (Achlioptas et al., 2018), PC-GAN (Li et al., 2018a), WarpingGAN (Tang et al., 2022), PointFlow (Yang et al., 2019), PVD (Zhou et al., 2021), DPC (Luo & Hu, 2021) and SetVAE (Kim et al., 2021). We adopt the popular evaluation metrics for point cloud generation, i.e., JSD, MMD-CD, MMD-EMD, COV-CD, COV-EMD (Achlioptas et al., 2018) and 1-NNA (Yang et al., 2019). JSD scores and MMD-EMD scores are multiplied by $10^2$. MMD-CD scores are multiplied by $10^3$. For JSD, MMD and 1-NNA, lower scores denote better performance. As shown in Table 1, our model outperforms raw-GAN across all three categories with a large margin and obtains either comparable or the best score under all evaluation metrics. In addition, we also show some qualitative results in Fig. 5.

## 5.3 The Evolution of Generation

In this subsection, we mainly unveil how these 2d patches are used to construct the final 3d shape. To better illustrate patch-wise generation, we plot the patches with different colors in Fig. 6, where three patch generators perform quite differently. Specifically, MLP-G generate each patch independently from the patch prior, and we find that there is little overlap between different generated patches. We have similar observation for PointTrans-G. Different from the above-mentioned two patch generators, DualTrans-G enhances the information flow between both points and patches, where each patch spreads over the surface of the 3d shape and produces uniformly distributed points.

## 5.4 Ablation Studies

**Number of Patches** We compare the generated results of three generators when using different numbers of patches and investigate how the number of patches influences the generation process. Taken the chair class for example, we show the MLP-G generation process of 2-patches and 4-patches in Fig. 7, where 2-patches and 4-patches generation do not fully construct the 3d shape, mainly because less patches cannot fit the

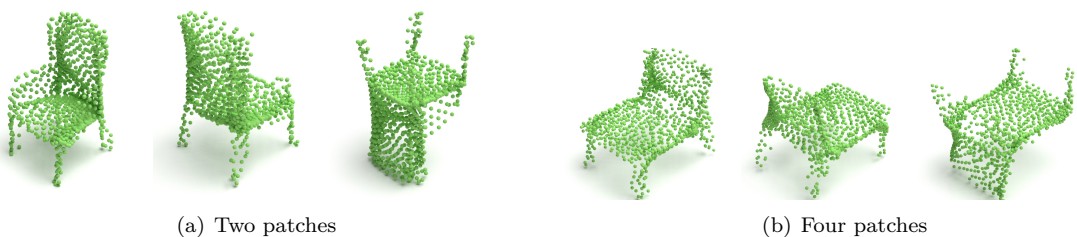

(a) Two patches        (b) Four patches

Figure 7: Point clouds generated by MLP-G using different patches (two patches and four patches) and observed from different views.

| Generator | MLP-G | | PointTrans-G | | DualTrans-G | |
|---|---|---|---|---|---|---|
| | Params | FLOPs | Params | FLOPs | Params | FLOPs |
| 2 patches | 60.52K | 58.7M | 1.44M | 2.68G | 93.55M | 0.60G |
| 4 patches | 116.94K | 58.7M | 2.48M | 1.87G | 85.12M | 1.04G |
| 8 patches | 229.78K | 58.7M | 4.57M | 1.47G | 80.91M | 1.92G |
| 16 patches | 455.46K | 58.7M | 8.73M | 1.27G | 78.81M | 3.67G |
| 32 patches | 906.82K | 58.7M | 17.07M | 1.17G | 77.77M | 7.19G |

Table 2: Model complexity of different patches.

surface of the 3d shape. However, with more patches, the complexity of the model will increase, and we also show the model complexity in Table 2. In practice, we find that 8-patches is a proper tradeoff in most cases.

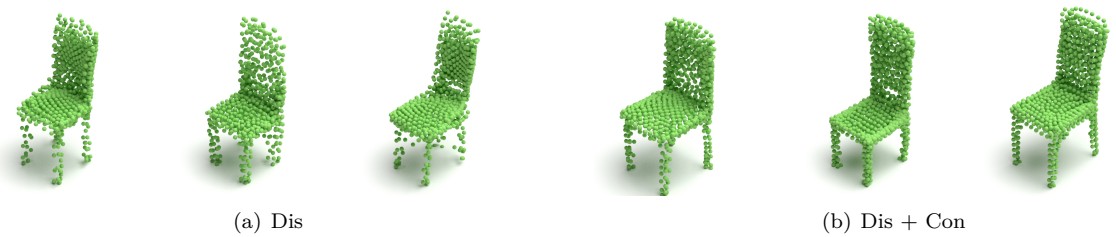

(a) Dis    (b) Dis + Con

Figure 8: Some examples of generated point clouds. The left (Dis) is trained without contrastive loss while the right (Dis+Con) is generated under the supervision of contrastive loss. From left to right: MLP-G, PointTrans-G, and DualTrans-G.

**Contrastive Loss**  In this work, we introduce the contrastive loss to further boost the generation process. Here, we give the qualitative and quantitative comparisons with results generated without the supervision of contrastive loss. Also taken the class chair for example, we present the quantitative results in Table 3 using the same evaluation metrics. Comparing with the results in Table 1, we see that contrastive loss benefits the generation quality a lot. In addition, we show some examples of generated point clouds in Fig. 8.

| Class | Model | JSD(↓) | MMD(↓) | | COV(↑, %) | | 1-NNA(↓, %) | |
|---|---|---|---|---|---|---|---|---|
| | | | CD | EMD | CD | EMD | CD | EMD |
| Chair | MLP-G | 2.14 \| 2.20 | 2.45 \| 2.49 | 7.83 \| 7.80 | 47.21 \| 47.09 | 46.52 \| 46.46 | 59.90 \| 59.94 | 60.31 \| 60.41 |
| | PointTrans-G | 1.92 \| 2.13 | 2.47 \| 2.55 | 7.75 \| 7.86 | 47.17 \| 47.15 | 46.37 \| 46.35 | 58.55 \| 58.61 | 60.74 \| 60.79 |
| | DualTrans-G | 1.87 \| 1.99 | 2.37 \| 2.38 | 7.69 \| 7.73 | 47.30 \| 47.26 | 46.63 \| 46.60 | 57.82 \| 57.88 | 60.06 \| 60.15 |
| Car | MLP-G | 0.83 \| 0.89 | 0.87 \| 0.87 | 5.17 \| 5.23 | 48.75 \| 48.70 | 46.81 \| 46.73 | 60.34 \| 60.47 | 61.95 \| 62.06 |
| | PointTrans-G | 0.80 \| 0.87 | 0.87 \| 0.88 | 5.24 \| 5.36 | 48.30 \| 48.28 | 46.22 \| 46.14 | 59.81 \| 59.89 | 62.26 \| 62.33 |
| | DualTrans-G | 0.78 \| 0.85 | 0.85 \| 0.90 | 5.14 \| 5.24 | 49.12 \| 49.10 | 46.54 \| 46.51 | 59.60 \| 59.63 | 61.81 \| 61.86 |
| Airplane | MLP-G | 4.58 \| 4.63 | 0.236 \| 0.241 | 3.04 \| 3.07 | 44.30 \| 44.25 | 49.10 \| 49.13 | 76.90 \| 76.86 | 76.11 \| 76.34 |
| | PointTrans-G | 4.57 \| 4.60 | 0.186 \| 0.188 | 3.12 \| 3.17 | 45.87 \| 45.82 | 48.58 \| 48.52 | 75.21 \| 75.25 | 75.40 \| 75.48 |
| | DualTrans-G | 4.61 \| 4.64 | 0.180 \| 0.179 | 3.04 \| 3.09 | 45.55 \| 45.49 | 49.22 \| 49.02 | 74.97 \| 75.06 | 75.31 \| 75.40 |

Table 3: Quantitative metrics with (left) and without (without) contrastive loss. The evaluation metrics are the same as Table 1.

## 5.5  Generation Process Probing

The key difference between our work and other methods is that we use multiple patches to generate point clouds. Thus, it is beneficial to explore the generation process that how these patches gradually construct the 3D shape. Taken the chair class for example, we demonstrate the results of three generator modules

at different epochs in Fig. 9. As expected, the three generator modules perform differently. MLP-G takes as input patches and deform them independently. The same situation also occurs in the generation of PointTrans-G. Different from that, DualTrans-G enhances the information flow between points and patches.

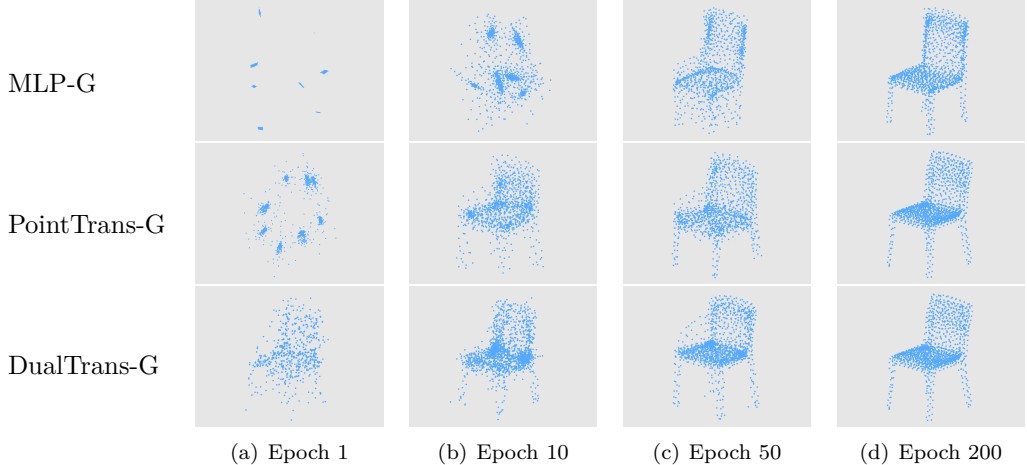

MLP-G

PointTrans-G

DualTrans-G

(a) Epoch 1      (b) Epoch 10      (c) Epoch 50      (d) Epoch 200

Figure 9: Examples of point clouds generated at different epochs. From left to right: epoch 1, epoch 10, epoch 50, and epoch 200.

### 5.6 Divide-and-Conquer Rethinking

As mentioned in Section 4.2, the pointwise reconstruction loss guarantees the final assembled point cloud to be satisfying. Therefore, it is helpful to unveil what the results of patch-wise generation would be like if there is no pointwise reconstruction loss. In our work, we adopt the VAE-GAN (Larsen et al., 2016) framework because it naturally incorporates the pointwise reconstruction loss. Here, we try to reproduce the generation using the GAN (Goodfellow et al., 2014) architecture and the loss is

$$\mathcal{L}_{GAN} = \mathbb{E}_{x \sim p_{data}}[log(D(x))] + \mathbb{E}_{z \sim p(z)}[log(1 - D(G(z)))], \tag{10}$$

where $p_{data}$ is the distribution of the training set and $p(z)$ is the prior distribution. $D$ is the discriminator and $G$ is the generator (or the decoder). For fair comparison, we reuse the decoder and the discriminator of VAE-GAN, as shown in Fig. 10. We keep all training hyper-parameters same as VAE-GAN. Taken MLP-G generator as an example, we show the generation result on the chair category in Fig. 11, where we find that, without the supervision of pointwise reconstruction loss, the results would be worse.

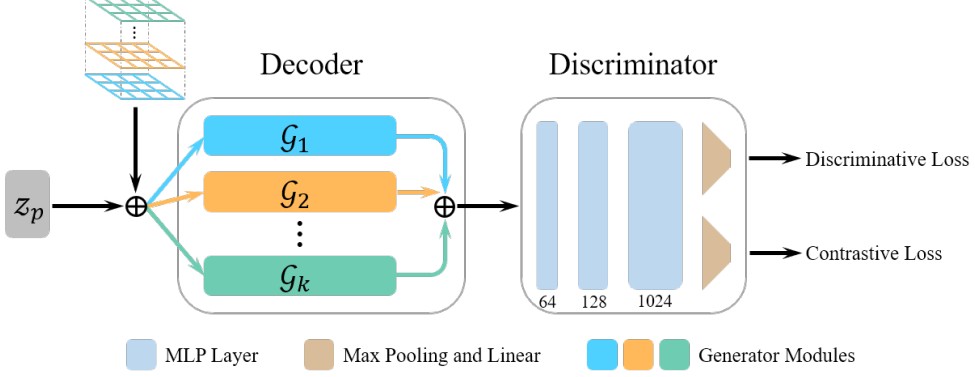

Figure 10: The overall GAN architecture with the same decoder and discriminator as the VAE-GAN.

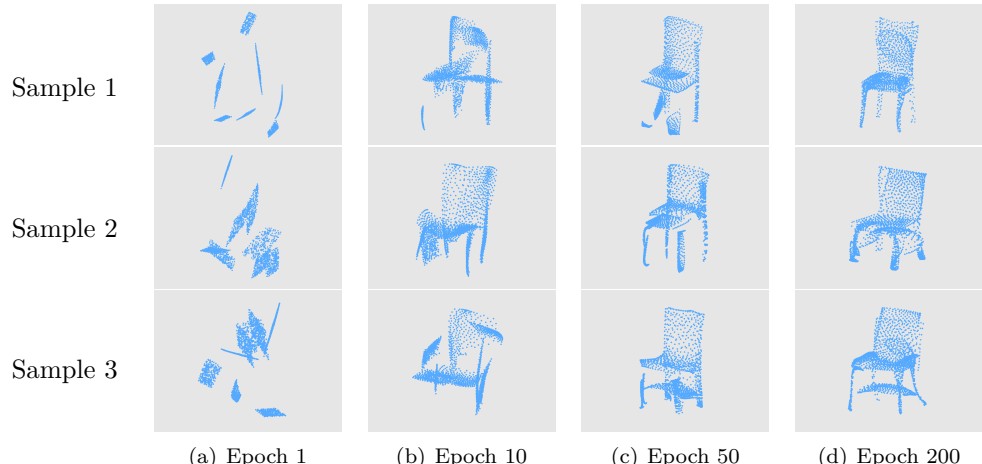

Figure 11: Examples of point cloud generated at different epochs using the GAN architecture. From left to right: epoch 1, epoch 10, epoch 50 and epoch 200.

| Model | MN10(%) | MN40(%) |
|---|---|---|
| 3D-GAN (Wu et al., 2016) | 91.0 | 83.3 |
| PointFlow (Yang et al., 2019) | 93.7 | 86.8 |
| PDGN (Hui et al., 2020a) | **94.2** | 87.3 |
| Ours | 94.0 | **87.4** |

Table 4: Classification results on ModelNet10 (MN10) and ModelNet40 (MN40).

### 5.7 Unsupervised Representation Learning

To further evaluate the proposed method for unsupervised representation learning, we conduct 3d object classification experiments as follows. Following previous works (Wu et al., 2016; Yang et al., 2019), we train our network on ShapeNet (Chang et al., 2015) and test it on two popular datasets, ModelNet10 (Wu et al., 2015) and ModelNet40 (Wu et al., 2015). Specifically, we feed our network with the full ShapeNet dataset and then extract the embedded features of the trained discriminator to learn a linear SVM for classification. We compare our network with recent state-of-the-art point cloud generation methods in terms of classification accuracy in Table 4. The results show that our method can extract discriminative features and thus generate high-quality 3d point cloud.

## 6 Conclusion

In this paper, we devise a novel patch-wise point cloud generation framework with three different generators: MLP-G, PointTrans-G and DualTrans-G. To generate realistic point clouds, we feed the generator with the concatenation of both latent representations and 2d patches. Extensive experiments show that the proposed method is able to produce high-fidelity point clouds and outperforms most recent point cloud generation methods under a variety of different evaluation metrics.

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

# A    Different Patches

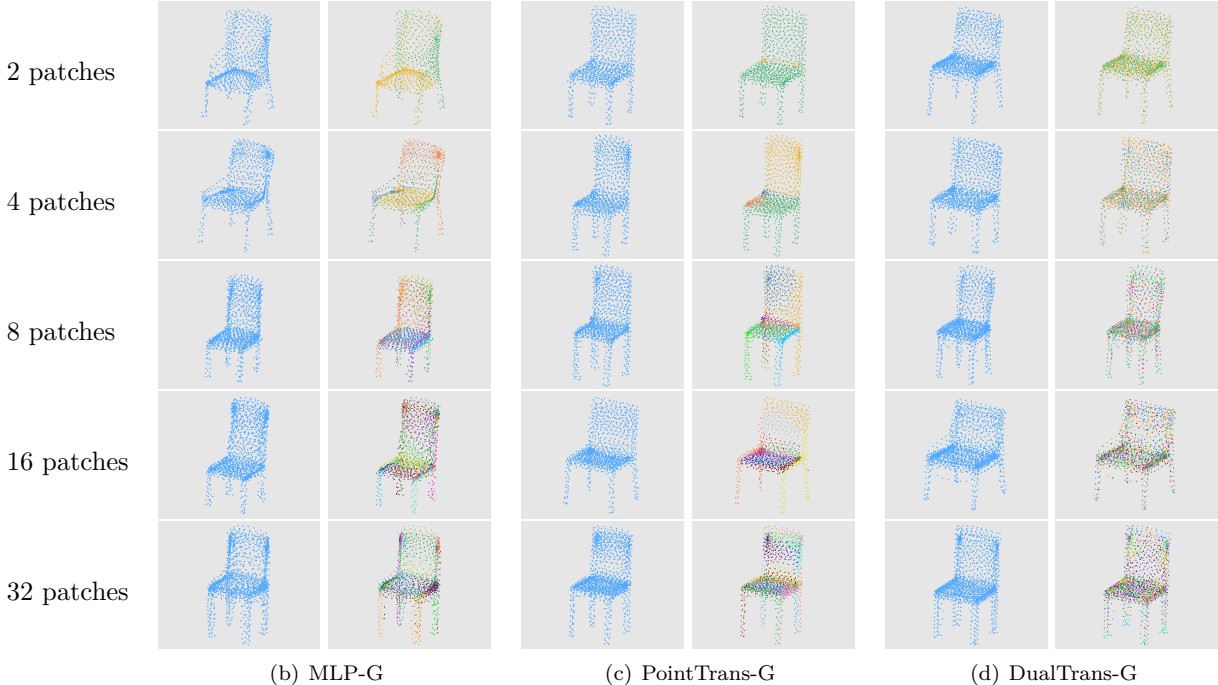

2 patches

4 patches

8 patches

16 patches

32 patches

(b) MLP-G          (c) PointTrans-G          (d) DualTrans-G

Figure 12: Examples of point cloud generated by our model with different patches.

Taken the chair class for example, in this section we present the gallery of point clouds generated with different patches and quantitative results in Table 5. As shown in Fig 12, for better view we plot the patches with different colors in the right column of each sub-figure. As mentioned in our main paper, we find that 8-patches is a proper tradeoff in most cases.

# B    Generation without Patches

All three variations of our model are based on patch-based generators and thus it is beneficial to show a direct comparison against a non-patch-based variant. Here, we present the qualitative and quantitative generation without patches in Fig 13 and Table 6. As shown in Fig 13, the main drawback of this method is that the points are not uniformly distributed.

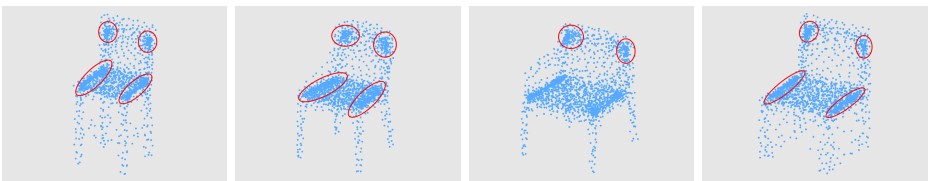

Figure 13: Examples of point cloud generated by VAE (without patches).

# C    Details in Our Network

In the main paper, we have shown the overall framework of our point cloud generative network. Here, we provide more details of the three generator modules. Note that, each type of transformer layer only contains one encoder module, thus we do not list the depth parameter in the following tables (Table 7, 8 and 9).

| Model | Patches | JSD(↓) | MMD(↓) | | COV(↑, %) | | 1-NNA(↓, %) | |
|---|---|---|---|---|---|---|---|---|
| | | | CD | EMD | CD | EMD | CD | EMD |
| MLP-G | 2 | 2.87 | 2.77 | 8.60 | 43.30 | 40.76 | 65.14 | 69.47 |
| | 4 | 2.59 | 2.62 | 8.14 | 45.55 | 44.35 | 62.13 | 67.59 |
| | 8 | 2.14 | 2.45 | 7.83 | 47.21 | 46.52 | 59.90 | 60.31 |
| | 16 | 2.05 | 2.45 | 7.92 | 47.53 | 46.31 | 59.84 | 61.17 |
| | 32 | 2.17 | 2.51 | 7.90 | 47.16 | 46.18 | 60.28 | 62.48 |
| PointTrans-G | 2 | 2.71 | 2.61 | 8.38 | 42.89 | 43.02 | 64.17 | 68.67 |
| | 4 | 2.23 | 2.59 | 8.02 | 44.56 | 45.44 | 62.64 | 65.54 |
| | 8 | 1.92 | 2.47 | 7.75 | 47.17 | 46.37 | 58.55 | 60.74 |
| | 16 | 1.84 | 2.44 | 7.70 | 47.42 | 46.61 | 58.97 | 61.45 |
| | 32 | 1.89 | 2.45 | 7.73 | 47.05 | 46.24 | 59.96 | 62.50 |
| DualTrans-G | 2 | 3.15 | 2.65 | 8.45 | 42.08 | 42.97 | 62.20 | 66.87 |
| | 4 | 2.02 | 2.51 | 8.34 | 45.84 | 45.76 | 60.43 | 64.33 |
| | 8 | 1.87 | 2.37 | 7.69 | 47.30 | 46.63 | 57.82 | 60.06 |
| | 16 | 1.89 | 2.33 | 7.72 | 47.21 | 46.68 | 58.73 | 61.13 |
| | 32 | 1.97 | 2.40 | 7.73 | 47.19 | 46.13 | 58.60 | 61.66 |

Table 5: Quantitative comparisons with different patches. The evaluation metrics are the same as our main paper.

| Model | JSD(↓) | MMD(↓) | | COV(↑, %) | | 1-NNA(↓, %) | |
|---|---|---|---|---|---|---|---|
| | | CD | EMD | CD | EMD | CD | EMD |
| VAE | 7.45 | 2.62 | 10.2 | 40.45 | 22.86 | 69.43 | 89.75 |
| MLP-G | 2.14 | 2.45 | 7.83 | 47.21 | 46.52 | 59.90 | 60.31 |
| PointTrans-G | 1.92 | 2.47 | 7.75 | 47.17 | 46.37 | 58.55 | 60.74 |
| DualTrans-G | 1.87 | 2.37 | 7.69 | 47.30 | 46.63 | 57.82 | 60.06 |

Table 6: Quantitative comparisons with none-patch generation. The evaluation metrics are the same as our main paper.

| DualTrans-G | Layer Type | In_dim | Out_dim | Heads |
|---|---|---|---|---|
| Layer 1 | pointwise transformer | 130 | 1024 | 8 |
| Layer 2 | patchwise transformer | 1024 | 1024 | 8 |
| Layer 3 | pointwise transformer | 1024 | 1024 | 8 |
| Layer 4 | patchwise transformer | 1024 | 1024 | 8 |
| Layer 5 | pointwise transformer | 1024 | 1024 | 8 |
| Layer 6 | patchwise transformer | 1024 | 1024 | 8 |
| Layer 7 | pointwise transformer | 1024 | 1024 | 8 |
| Layer 8 | patchwise transformer | 1024 | 3 | 8 |

Table 9: Details of DualTrans-G.

# D Model Complexity Analysis

In Subsection 5.4 of the main paper, we have given the model complexity with different input patches. Here, we discuss the parameters and FLOPs in details. For convenience, we paste the model complexity

| MLP-G | Layer Type | In_dim | Out_dim |
|---|---|---|---|
| Layer 1 | MLP | 130 | 128 |
| Layer 2 | MLP | 128 | 64 |
| Layer 3 | MLP | 64 | 32 |
| Layer 4 | MLP | 32 | 3 |

Table 7: Details of MLP-G.

| PointTrans-G | Layer Type | In_dim | Out_dim | Heads |
|---|---|---|---|---|
| Layer 1 | MLP | 130 | 128 | - |
| Layer 2 | pointwise transformer | 128 | 128 | 4 |
| Layer 3 | pointwise transformer | 128 | 128 | 4 |
| Layer 4 | MLP | 128 | 64 | - |
| Layer 5 | pointwise transformer | 64 | 64 | 4 |
| Layer 6 | pointwise transformer | 64 | 64 | 4 |
| Layer 7 | MLP | 64 | 3 | - |

Table 8: Details of PointTrans-G.

table of our main paper here as Table 10. Before we analyse the model complexity, we first illustrate two core components of the transformer layer, i.e., the self-attention module and the position-wise feed-forward network (FFN) in Table 11 (Lin et al., 2021).

| Generator | MLP-G | | PointTrans-G | | DualTrans-G | |
|---|---|---|---|---|---|---|
| | Params | FLOPs | Params | FLOPs | Params | FLOPs |
| 2 patches | 60.52K | 58.7M | 1.44M | 2.68G | 93.55M | 0.60G |
| 4 patches | 116.94K | 58.7M | 2.48M | 1.87G | 85.12M | 1.04G |
| 8 patches | 229.78K | 58.7M | 4.57M | 1.47G | 80.91M | 1.92G |
| 16 patches | 455.46K | 58.7M | 8.73M | 1.27G | 78.81M | 3.67G |
| 32 patches | 906.82K | 58.7M | 17.07M | 1.17G | 77.77M | 7.19G |

Table 10: Model complexity of different patches.

**MLP-G**   In this generator module, when the number of patches increases, the Params will increase because each patch associates to a MLP generator module. Although the FLOPs of each MLP generator module decrease, the total number of points (2048 in our experiments) keep fixed, and thus the total FLOPs keep the same.

**PointTrans-G**   The parameters in PointTrans-G will increase with more input patches. Although the FLOPs of the MLP layers keep the same, the FLOPs of pointwise transformer layer decrease because of the $O(T^2 \cdot D)$ complexity in Table 11.

**DualTrans-G**   The DualTrans-G is different from the previous two generator modules because all patches are fed in simultaneously. The output of the last patchwise transformer layer is point clouds with size $N_k \times 3$, where $N_k$ is the number of points on current patch. When the number of patches increases, $N_k$ decreases and then leads to small Params. In our settings, the input and output channels of transformer layers are fixed. Therefore, when the number of patches doubles, the overall FLOPs will increase.

**Self-Attention Heads**   The number of transformer layers and the self-attention heads in each type of transformer are also important hyper-parameters of Point-Trans-G and DualTrans-G. Here, we give the

| Module | Parameters | Complexity |
|---|---|---|
| self-attention | $4D^2$ | $O(T^2 \cdot D)$ |
| position-wise FFN | $8D^2$ | $O(T \cdot D)$ |

Table 11: Parameters and complexity of self-attention and position-wise FFN. $D$ is the hidden dimension and $T$ is the input sequence length. The intermediate dimension of FFN is set to $4D$.

model complexity when using different parameters, shown in Table 12 and Table 13. In our experiments, these parameters does not have a huge influence on the final generated results, but on the model parameters and training time.

| PointTrans-G | Num | Heads | Params | FLOPs |
|---|---|---|---|---|
| | 2 | 6 | 4.53M | 1.45G |
| | | 8 | 4.57M | 1.47G |
| pointwise transformer | 4 | 4 | 8.53M | 2.89G |
| | | 6 | 8.46M | 2.85G |
| | | 8 | 8.53M | 2.89G |
| | 6 | 4 | 12.49M | 4.30G |
| | | 6 | 12.39M | 4.24G |
| | | 8 | 12.49M | 4.30G |

Table 12: PointTrans-G complexity of different transformer settings. Num is the number of pointwise transformer layers between MLP layers and Heads is the number of self-attention heads in the transformer.

| DualTrans-G | Num | Heads | Params | FLOPs |
|---|---|---|---|---|
| | 4 | 6 | 80.84M | 1.92G |
| | | 10 | 80.84M | 1.92G |
| pointwise transformer | 6 | 6 | 0.13G | 3.06G |
| or | | 8 | 0.13G | 3.07G |
| patchwise transformer | | 10 | 0.13G | 3.06G |
| | 8 | 6 | 0.18G | 4.21G |
| | | 8 | 0.18G | 4.21G |
| | | 10 | 0.18G | 4.21G |

Table 13: DualTrans-G complexity of different transformer settings. Num is the number of transformer layers and Heads is the number of self-attention heads in each transformer. Num and Heads apply to both transformers. For example, Num=4 and Heads=6 mean that there are 4 pointwise transformer layers and 4 patchwise transformer layers in DualTrans-G, and each transformer have 6 self-attention heads.

# E  More Results

In this section, we show more generated results of our method in Fig. 15, and for comparison, we also present some examples of raw-GAN (Achlioptas et al., 2018), PointFlow (Yang et al., 2019) and SetVAE (Kim et al., 2021) in Fig. 14.

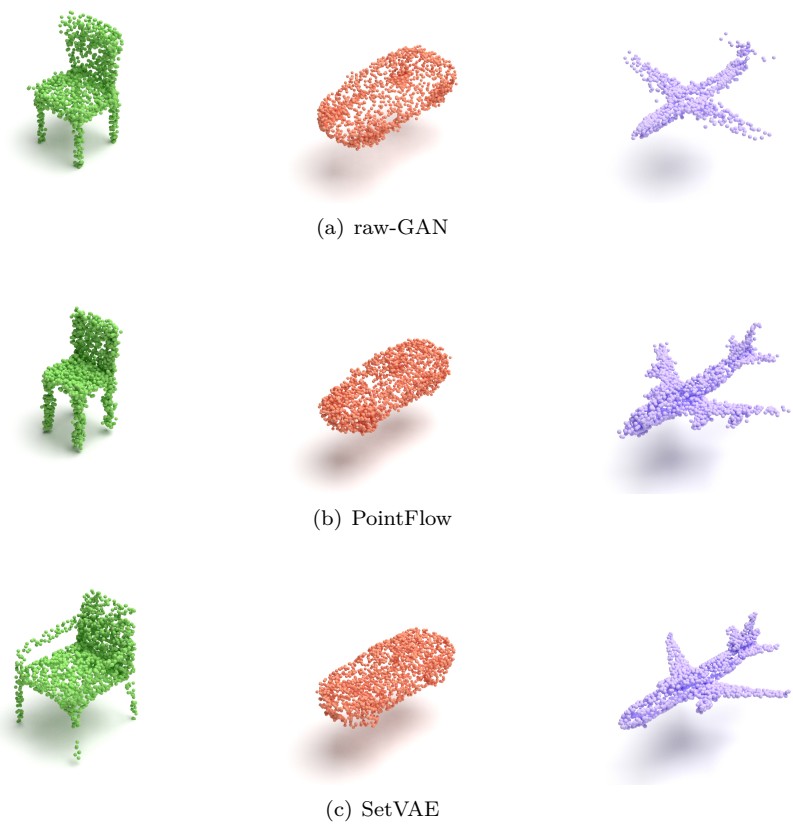

(a) raw-GAN

(b) PointFlow

(c) SetVAE

Figure 14: Examples of point clouds generated by raw-GAN (Achlioptas et al., 2018), PointFlow (Yang et al., 2019) and SetVAE (Kim et al., 2021).

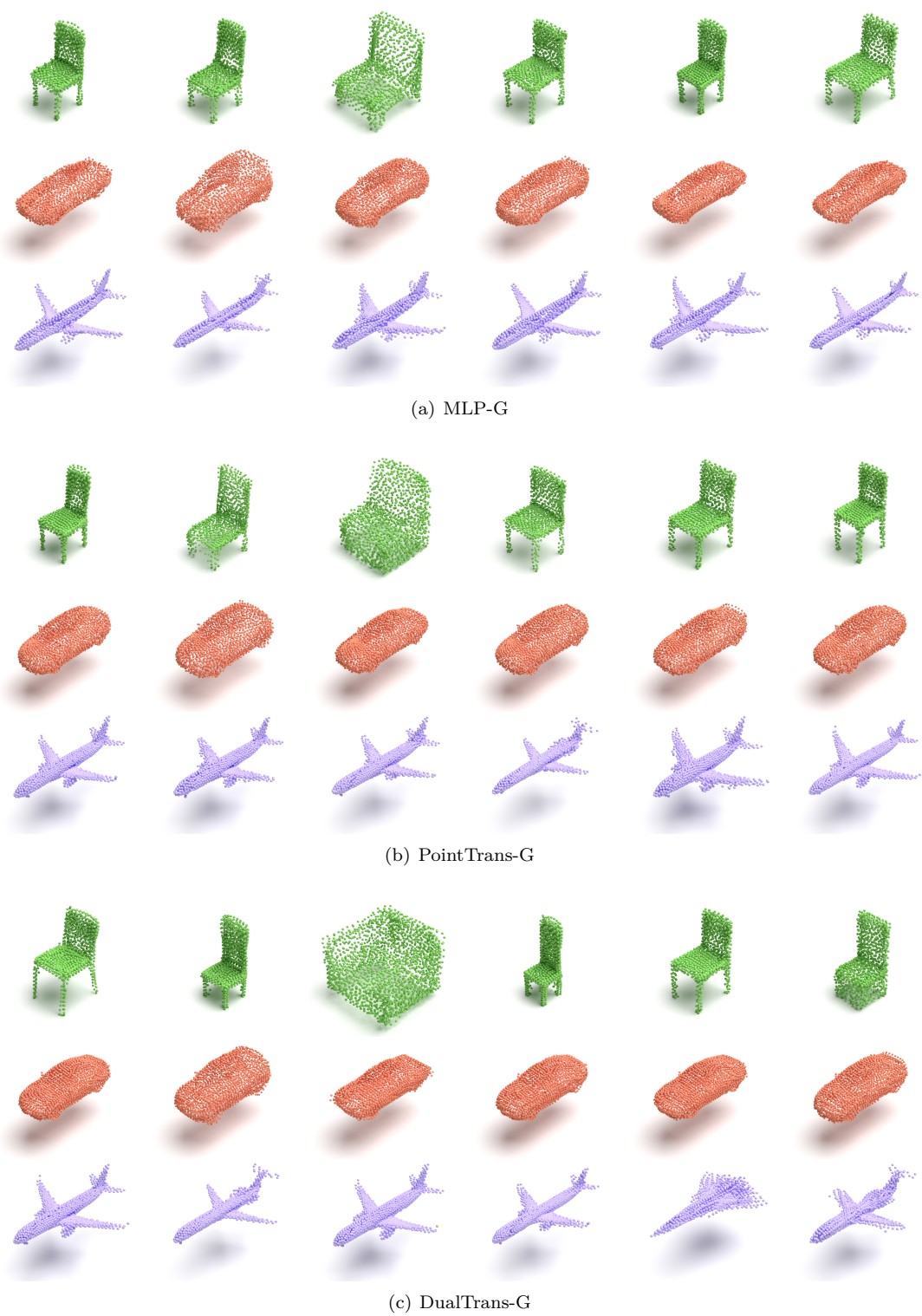

(a) MLP-G

(b) PointTrans-G

(c) DualTrans-G

Figure 15: Examples of point clouds generated by our model. From top to bottom: chair, car and airplane.

