# OpenReview forum: "Patch-Wise Point Cloud Generation: A Divide-and-Conquer Approach"
_TMLR — Rejected by TMLR_

### Review · Reviewer_EQjC · 2022-08-10

**Summary Of Contributions:**

The paper introduced a new point cloud generation method -- to generate the content, the algorithm was fed with 2D patches and latent representations and the generated object parts are assembled to formulate the entire object. The authors introduced 3 new algorithms for the generators with different variants, and demonstrated decent performances when comparing with concurrent algorithms like SetVAE, PointFlow, etc. The authors also did some ablation studies for their proposed variants and discussed some performance trade-offs.

**Broader Impact Concerns:**

It's perhaps useful to discuss potential broader impact factors for the generated contents -- though only point cloud is generated, it may be advised to use the algorithm with caution when trying to generate new contents for new categories.

**Requested Changes:**

I'm not sure if accepting the paper can be justified without seeing more up-to-date comparisons with most concurrent works (see weaknesses) and it will be good if authors can address other raised questions too. From the algorithm's perspective, I'm more interested in seeing the motivations of b+c in Fig.4, as they are relying on well-studied transformer architectures and not introducing new architectures.

**Strengths And Weaknesses:**

[Strengths]
- The overall presentation of this paper is good. I like the visualization/background sections and the writing of the paper -- they help me better understand the formulations of the paper
- The new algorithm seems to be performing well from visualizations and evaluations on benchmarks. The formulations of the new algorithm is sensible as well.

[Weaknesses]
- In Sec 4.2, the authors uses a symmetric chamfer distance and cited [Fan et al, 2017], however to my understanding that paper introduced chamfer distance loss and I'm not sure what's the correct definition of symmetric CD.
- In Fig. 4 as well as Sec 4.3, the authors introduced new network architectures b+c of using transformers for this generative model. However I don't fully get the motivation of the approach -- is it more on accuracies or on efficiencies? Why is that useful besides improvements on evaluation metrics?
- In Table 1, it looks like the proposed method not always out-perfrom other methods. For example on chairs -- is there any analysis or visual comparisons for failure cases?
- Table 2 shows #patches impact the computational costs, but does more patch leads to better accuracy?
- The proposed method is compared with a few works up to 2021, but there are many related works, such as "Attention-based Transformation from Latent Features to Point Clouds, AAAI 2022", "WarpingGAN: Warping Multiple Uniform Priors for Adversarial 3D Point Cloud Generation, CVPR 2022" (cited but not compared), "Diffusion Probabilistic Models for 3D Point Cloud Generation, CVPR 2021" (cited but not compared). I wonder if the conclusion is still up-to-date.

---

> ### Author Response · Authors · 2022-09-06
> **Answer to Reviewer EQjC**
>
> **Q1**: ``In Sec 4.2, the authors use a symmetric chamfer distance and cited [Fan et al, 2017], however to my understanding that paper introduced chamfer distance loss and I'm not sure what's the correct definition of symmetric CD.''
> **A1**: We follow the usage as [1]. Actually, symmetric CD should be the same as CD defined in [Fan et al, 2017].
> **Refs**:
> $[1]$ Learning elementary structures for 3d shape generation and matching.
>
> **Q2**: ``In Fig. 4 as well as Sec 4.3, the authors introduced new network architectures b+c of using transformers for this generative model. However, I don't fully get the motivation of the approach -- is it more on accuracies or on efficiencies? Why is that useful besides improvements on evaluation metrics?''
> **A2**: To better understand this, we take PointNet as an example. PointNet always extracts features in a point-wise but isolated manner, that is, there is no information flow between local points, leading to inferior performance in point cloud classification and segmentation comparing with following works such as PointNet++ and PointCNN. In our work, the same situation also occurs for MLP generation (patch-level information block) and PointTrans generation (point-level information block). Inspired by recent success of vision transformers, we introduce the transformer structure to extract local features by aggregating the information of different patches and points.
>
> **Q3**: ``In Table 1, it looks like the proposed method not always out-perfrom other methods. For example on chairs -- is there any analysis or visual comparisons for failure cases?
> Table 2 shows patches impact the computational costs, but does more patch leads to better accuracy?''
> **A3**: We will provide some failure cases in the project page in future and also discuss possible reasons. As shown in Section A of Appendix, more patches does not always lead to a better accuracy. In our paper, we find that 8-patches is a proper tradeoff in most cases.
>
> **Q4**: ``The proposed method is compared with a few works up to 2021, but there are many related works, such as "Attention-based Transformation from Latent Features to Point Clouds, AAAI 2022", "WarpingGAN: Warping Multiple Uniform Priors for Adversarial 3D Point Cloud Generation, CVPR 2022" (cited but not compared), "Diffusion Probabilistic Models for 3D Point Cloud Generation, CVPR 2021" (cited but not compared). I wonder if the conclusion is still up-to-date.''
> **A4**: We have included the comparisons with most recent works in Table 1 of the revised version.

---

### Review · Reviewer_bJ5e · 2022-08-10

**Summary Of Contributions:**

This paper presents a patch-based point cloud generation model. The key idea is simply to generate the point clouds in a set of point cloud patches. The model is trained in a VAE-GAN framework, with an additionally introduced instance contrastive loss taking augmented (eg, rotated) point cloud as positive sample and different point cloud instance as negative sample. The paper presents experiments with three different generator architectures, including simple point-wise MLP, point-level transformer and patch-level transformer, and the results show the patch-level transformer architecture performs slightly better. Overall, the generation results seem to slightly better than existing methods on ShapeNet objects (chair, car and airplane).

**Broader Impact Concerns:**

As with all generative models, there is a risk of misuses in deepfake applications. However, I think the ethical concerns here on generating point clouds are less significant than generating real images.

**Requested Changes:**

- (Critical) Please highlight the major technical insights and explain why they are important.
- Polishing the writing throughout the paper by emphasizing on the motivation and insights would certainly significantly strengthen the manuscript.

**Strengths And Weaknesses:**

## Strengths

### S1 - Patch-based point cloud generation seems new
- The patch-based generation pipeline is similar to AtlasNet applied to point clouds. Despite being simple and intuitive, I cannot think of any prior work adopts similar approach for point cloud generation.
- The instance contrastive loss also seems interesting. Visually in Fig. 8, it does seem to improve the overall fidelity of the generated point clouds, although quantitatively, I do not find the performance improvement very significant, by comparing Table 3 and Table 1.

### S2 - Competitive results on a standard benchmark
The experiment results compare favorably against SOTA point cloud generation methods on ShapeNet objects.

## Weaknesses

### W1 - Not much technical insight
The key insight of using a patch-based generator for point clouds is pretty intuitive and the results seem reasonable with different generator architectures. But overall nether the method nor the result seems very exciting.

### W2 - Poor presentation
The lack of technical insight can also be attributed to the poor presentation. Overall, the paper consists mostly of dry descriptions of the method and the results. It is not clear what technical insights can be drawn from these results.

There are also several other issues through the paper. For examples:
- The term "patch prior" is very confusing. If I understand correctly, all these so-called "patch priors" $\mathcal{P}_i$ are just a fixed grid of 2D pixel coordinates (eg, a 2D grid from -1 to 1). These coordinates themselves do not capture any prior at all. What actually captures the priors is the generators $\mathcal{G}_i$. This causes a lot of confusion throughout the paper, eg "the learnable 2d patch priors are used to generate point cloud patches", "We always use two dimensional patch priors, i.e., $d_\text{patch} = 2$".
- I would suggest not to overload the notations to take in both samples $x$ and noise $z$ as input to avoid confusion. Eg, in Eq (3): $F_D(x)$ vs. $F_D(z)$. I suppose $F_D(z)$ is meant to be $F_D(G(z))$. Similarly for $D(x)$ vs. $D(z)$ in Eq (4).
- Table 1: explain all the metrics in the texts.
- Table 3: put the results of the full model and the ablated model side by side for easy comparison.
- Sec 5.6: the section is titled "Divide-and-Conquer Rethinking" whereas the content is about the ablation without the VAE (with only the GAN loss). I don't understand what the ablation results reveal about the "divide-and-conquer" strategy.

### W3 - Ablation on the patch-based generation
As the key contribution is the patch-based generator, a thorough comparison between non-patch-based counterpart is expected. If I interpret the results correctly, all three variations of the model are based on patch-based generators and none of the experiments show a direct comparison against a non-patch-based variant (except for other existing methods).

---

> ### Author Response · Authors · 2022-09-06
> **Answer to Reviewer bJ5e**
>
> **Q1**: ``Not much technical insight. The key insight of using a patch-based generator for point clouds is pretty intuitive and the results seem reasonable with different generator architectures. But overall neither the method nor the result seems very exciting.
> Poor presentation. The lack of technical insight can also be attributed to the poor presentation. Overall, the paper consists mostly of dry descriptions of the method and the results. It is not clear what technical insights can be drawn from these results.''
> **A1**: Our motivation is that an image can be divided into 2D patches and similarly point cloud can also be divided into 3D patches. Basically, we have two key  insights: 1) From the generation view, the divide-and-conquer generation can be treated as the counterpart of progressive generation. The former generation is from part to whole, while the latter generation is from coarse to fine; 2) From the geometric view, the low dimension prior always contributes to high dimension generation, that is from 2D patches to 3D point clouds. In addition, our work is also an attempt that embeds 2D manifold into 3D space.
>
> **Q2**: ``The term "patch prior" is very confusing. If I understand correctly, all these so-called "patch priors" are just a fixed grid of 2D pixel coordinates (eg, a 2D grid from -1 to 1). These coordinates themselves do not capture any prior at all. What actually captures the priors is the generators...''
> **A2**: There are two distributions introduced in VAE, prior distribution $p(z)$ (always the normal distribution) and conditional likelihood distribution $p(z|x)$ where $x$ is the input. In our work, since the patch is always initialized by normal distribution, we call it patch prior, which is the counterpart of $p(z)$.
>
> **Q3**: ``I would suggest not to overload the notations to take in both samples x and noise z as input to avoid confusion. Eg, in Eq (3): $F_D(x)$ vs. $F_D(z)$. I suppose $F_D(z)$ is meant to be $F_D(G(z))$. Similarly for $D(x)$ vs $D(z)$ in Eq (4).''
> **A3**: We have carefully revised this to avoid confusion in our revision. Please refer to Eq (3) and Eq (4).
>
> **Q4**: ``Table 1: explain all the metrics in the texts.''
> **A4**: We have revised this in our paper. Please refer to the descriptions in Page 9.
>
> **Q5**: ``Table 3: put the results of the full model and the ablated model side by side for easy comparison.''
> **A5**: We have revised this in our paper. Please refer to Table 3 in our revision.
>
> **Q6**: ``Sec 5.6: the section is titled "Divide-and-Conquer Rethinking" whereas the content is about the ablation without the VAE (with only the GAN loss). I don't understand what the ablation results reveal about the "divide-and-conquer" strategy.''
> **A6**: Just as its name implies, the "divide-and-conquer" strategy consists of two steps, dividing and conquering. To achieve the conquering step, we try to figure out whether the reconstruction loss is indispensable or not. The experiment in Sec 5.6 reveals that patch-wise generation is only applicable to framework with reconstruction loss, such as VAE-GAN but not GAN.
>
> **Q7**: ``Ablation on the patch-based generation. As the key contribution is the patch-based generator, a thorough comparison between non-patch-based counterpart is expected. If I interpret the results correctly, all three variations of the model are based on patch-based generators and none of the experiments show a direct comparison against a non-patch-based variant (except for other existing methods).''
> **A7**: We have added this experiment in our paper. Please refer to Section B in Appendix.

---

### Review · Reviewer_4DTV · 2022-08-23

**Summary Of Contributions:**

The paper proposes a generative point cloud method based on VAE-GAN with transformer-based decoders / generators and a contrastive loss based on discriminator features. A key idea is to split the shape generation process into multiple path-wise tasks. ShapeNet experiments show good performance compared to baselines.

**Broader Impact Concerns:**

looks good

**Requested Changes:**

I listed several requests in weaknesses.

**Strengths And Weaknesses:**

**Strengths:**
(+) The paper tackles an important task of pointcloud generation. writing is clear and easy to follow.
(+) A divide and conquer approach is suggested in order to shift mapping capacity from the generator to patch priors. Conceptually, this is a powerful idea.
(+) The authors discuss and experiment with an interesting design space of transformer-based generation architectures that operates at point- and patch- levels.


**Weaknesses:**
(-) Missing an important comparison with SoTA. Please see ref [1] and works it compares against in table 2. In particular, [1] improves upon point flow in 1-NNA CD between 4-6 points which should be quite competitive to the suggested method and hence an important baseline to include.

(-) Figure (6) concerns me. Observing the patches generated from the top performing DualTrans architecture, they are very overlapping. Instead of easing the generator's job by localizing the mapping, all mappings cover the entire space, which contrasts with divide and conquer motivation. In larger, more complex scenes, this may not scale well.

(-) Although I generally agree that the work differs from Groueix et al. in that its application is generation rather than reconstruction, since Groueix et al., is so close in design to the suggested method, and as the model is based on a VAE at its core, it would be very informative to replace the VAE with an AE and compare its reconstruction performance to Groueix's. We would be able to determine, for example, how important discriminator and contrastive losses are for reconstruction quality.

(-) While I appreciate the author's explanation, I'm still not sure I understand the need for a contrastive loss. I am especially puzzled by this part: "The contrastive loss encourages each instance to be different from the other samples in the sample batch, leading to high-fidelity generation." Why would diversity encourage fidelity?

(-) A total of three architectures are evaluated. Global, point-level, and combined patch and point. I’m missing one that only communicates at the patch level but not at the point-level. I am particularly curious about that since recent literature indicates that a single coordinate network could perform complex mapping. Therefore, point-level communication may be redundant. Nevertheless, patch-level makes a lot of sense to me since it breaks down complex geometry into smaller chunks. It would be great if the authors could complete the ablation and test a version with only patch level transformers.

(-) Do the different patch decoders in (a) and (b) share weights?

(-) How come the number of parameters decreases with the increase in number of patches for the DualTrans-G architecture in table 2?

(-) An analysis of the number of patches should also include a quantitative comparison of performance, rather than only a qualitative assessment of two examples. In addition, it would be nice to have a visual comparison of all patch numbers on the same encoded shape.

(-) Contrastive loss seems to matter only little according to the ablation study in terms of overall performance, except in the JSD. Analyses and visualizations would be helpful. Also, it seems that augmentations were introduced. For fair comparison, I’m curious if these augmentations are introduced during baseline training as well?

(-) Could the method work across categories or has to be trained per-category?


**Typos:**
(-) Page 6: "aims (to) fool”
(-) What’s Dis in eq (5)? Or is this a typo and should be D?


**Refs:**
[1] 3D Shape Generation and Completion through Point-Voxel Diffusion

---

> ### Author Response · Authors · 2022-09-06
> **Answer to Reviewer 4DTV**
>
> **Q1**: ``Missing an important comparison with SoTA. Please see ref [1] and works it compares against in table 2. In particular, [1] improves upon point flow in 1-NNA CD between 4-6 points which should be ...''
> **A1**: We have added the comparison with SoTA such as PVD [1]. Specifically, to compute the JSD, MMD-CD, MMD-EMD, COV-CD and COV-EMD, we have reproduced PVD~[1] and included the results in Table 1 of the revised version.
> **Refs**:
> $[1]$ 3D Shape Generation and Completion through Point-Voxel Diffusion.
>
> **Q2**: ``Figure (6) concerns me. Observing the patches generated from the top performing DualTrans architecture, they are very overlapping ...''
> **A2**: We would like to clarify that the divide-and-conquer solution not only means block by block, but also set by set where each set is downsampled from the shape. This is the key insight behind the dual transformer structure. Specifically, the outputs of MLP-G and PointTrans-G are divided into blocks and then assembled together. By contrast, the output of DualTrans-G is divided into subsets of the shape and then assembled, which also follows the code of divide-and-conquer and somewhat works like super-resolution.
>
> **Q3**: ``Although I generally agree that the work differs from Groueix et al. in that its application is generation rather than reconstruction, since Groueix et al., is so close in design to ...''
> **A3**: We would like to clarify that although both named reconstruction, it is not very good to directly compare between VAE and AE, since VAE introduces a Kullback-Leibler (KL) divergence term that acts like a regularizer on the reconstruction, which often creates issues due to the trade-off with prior distribution.
>
> **Q4**:  ``While I appreciate the author's explanation, I'm still not sure I understand the need for a contrastive loss. I am especially puzzled by this part: "The contrastive loss encourages each instance to be different from ...''
> **A4**: A key advantage of the contrastive loss over the standard reconstruction loss is its relaxed and instance-aware formulation. Basically, the reconstruction loss wants a perfect match between the reconstruction and the input, whereas the contrastive loss requests for being the most similar one among the training samples. This way, the contrastive loss becomes more cooperative with less conflict to the GAN loss, compared with the reconstruction loss. Therefore, the contrastive loss brings additional improvements over VAE.
>
> **Q5**: ``A total of three architectures are evaluated. Global, point-level, and combined patch and point. I’m missing one that only communicates at the patch level but not at the point-level ...''
> **A5**: To better understand this, we give a comparison between the convolutions on images and point clouds. For image tasks, we always use a filter kernel with receptive field $m \times n$ to extract local features. By contrast, for point cloud tasks, most works such as PointNet and DGCNN extract features in a point-wise manner, because each point scatters in the 3D space with its unique information. When coming to our work, we consider that point-level information is also of great importance for generation. That is also the reason why the dual structure is necessary.
>
> **Q6**: ``Do the different patch decoders in (a) and (b) share weights?''
> **A6**: No. The different patch decoders in (a) and (b) do not share weights.
>
> **Q7**: ``How come the number of parameters decreases ...''
> **A7**: We would like to refer the reviewer to the Section D in Appendix for a detail discussion on the model complexity analysis.
>
> **Q8**: ``An analysis of the number of patches should also include a quantitative comparison of performance, rather than only a qualitative assessment of two examples ...''
> **A8**: We have added this experiment in our revision. Please refer to Section A in Appendix.
>
> **Q9**: ``Contrastive loss seems to matter only little according to the ablation study in terms of overall performance, except in the JSD. Analyses and visualizations would be helpful ...''
> **A9**: As mentioned in PointFlow, all these popular metrics sometimes cannot ensure a fair model comparison for point cloud generation. In our experiment, we find there is performance gap between the numerical metrics and visual rendering, and the contrastive loss is quite useful in point cloud generation. We will give more analysis on this. Data preprocessing is applied to all methods, following from PointFlow.
>
> **Q10**: ``Could the method work across categories or has to be trained per-category?''
> **A10**: Generally, recent point cloud generation methods are devised to be trained per-category.
>
> **Q11**: ``Typo: Page 6: aims (to) fool''
> **A11**: We have carefully revised our paper to avoid this.
>
> **Q12**: ``Typo: What’s Dis in eq (5)? Or is this a typo and should be D?''
> **A12**: It is a typo and the $Dis$ should be $D$. We have corrected this in our paper. Please refer to Eq (5) in Page 4.

---

### Decision · Action_Editors · 2022-09-30

**Recommendation:** Reject

**Comment:**

This submission was reviewed by three expert reviewers.  After reading the revision and discussing the submission internally, all three reviewers leaned toward rejection.  Some reviewers are worried about the significance of the novelty.  Due to the nature of TMLR, this should not be a concern against the paper and such claims were not considered by the AE.  However, reviewer 4DTV provided a thoughtful summary of issues that remain in the revision.  This includes major presentation issues as well as the appropriateness of some performance claims. The AE agrees.  As these issues are major, the AE has to recommend rejection.  The authors are welcome to revise the paper to address these issues and resubmit.

Below are the comments from reviewer 4DTV.

**Methodology**:

(1) the main point of the work is to divide and conquer so that the shape is generated from patches. Yet, the strongest architecture presented does not seem to work in patches. The authors responded to this concern saying that the dual-transformer "somewhat works like super-resolution". I don't see why that would be the case given that the generators work in parallel. Also, the authors emphasize in their response to bj5e that their method "can be treated as the counterpart of progressive generation... is from part to whole, while the latter generation is from coarse to fine". This concerns me as this suggests that it is not the division into patches that makes the method work well and if that's the case further analysis and serious rewriting is needed. **This is my major concern and i'm leaning rejection of the current state of the manuscript based on that.**

(2) As was pointed out by me and other reviewers, the similarity to AtlasNet is large. I agree that comparing a VAE to an AE is not fair, but changing the suggested method to AE would help appreciate the contribution of the suggested generators. Namely, do the transformers improve **reconstruction** fidelity over a MLP?

(3) I do like the contrastive-loss and appreciate the comparison added to table 3. Yet, It would be great to know if adding similar augmentation (rotation and jitter) to the ELBO+GAN loss would't suffice? Although i asked that in my original response, i don't undertand from the answer: "Data preprocessing is applied to all methods, following from PointFlow" if the authors refer also to augmentations used?

**Results**: with the addition of the baselines suggested by me and the other reviewers, the suggested method no longer "clearly outperforms” baselines as suggested in the abstract. Yet, it is competitive. I would advise to tone down the claims. Additionally, I'm not sure why the reproduced results of PVD are much weaker than the reported in the original manuscript. Further, Is 64.83 for DPC in 1-NNA CD column for the airplane category a typo? Or is this really performing 10 its better than competitors?

**Audience:**

Yes

**Claims And Evidence:**

Some novelty and performance claims are not fully supported. Please see detailed comments below.